# DeCo-DETR: Decoupled Cognition DETR for Efficient Open-Vocabulary Object Detection

**Siheng Wang[1,4*], Yanshu Li[2*], Bohan Hu[4*], Zhengdao Li[4], Haibo Zhan[1], Linshan Li[1]**
**Weiming Liu[3], Ruizhi Qian[6], Guangxin Wu[6], Hao Zhang[6], Jifeng Shen[1†], Piotr Koniusz[5,8]**
**Zhengtao Yao[6†], Junhao Dong[3†], Qiang Sun[4,7 †]**

[1] Jiangsu University [2] Brown University [3] Nanyang Technological University [4] MBZUAI
[5] University of New South Wales [6] USC [7] University of Toronto [8] Data61♥CSIRO
`zyao9248@usc.edu`

## Abstract

Open-vocabulary object detection (OVOD) enables models to recognize objects beyond predefined categories, but existing approaches remain limited in practical deployment. On the one hand, multimodal designs often incur substantial computational overhead due to their reliance on text encoders at inference time. On the other hand, tightly coupled training objectives introduce a trade-off between closed-set detection accuracy and open-world generalization. Thus, we propose Decoupled Cognition DETR (DeCo-DETR), a vision-centric framework that addresses these challenges through a unified decoupling paradigm. Instead of depending on online text encoding, DeCo-DETR constructs a hierarchical semantic prototype space from region-level descriptions generated by pre-trained LVLMs and aligned via CLIP, enabling efficient and reusable semantic representation. Building upon this representation, the framework further disentangles semantic reasoning from localization through a decoupled training strategy, which separates alignment and detection into parallel optimization streams. Extensive experiments on standard OVOD benchmarks demonstrate that DeCo-DETR achieves competitive zero-shot detection performance while significantly improving inference efficiency. These results highlight the effectiveness of decoupling semantic cognition from detection, offering a practical direction for scalable OVOD systems.

## 1 Introduction

Open-vocabulary object detection (OVOD) enables inference-time localization and classification of both seen and unseen classes during training, thus overcoming the out-of-distribution limitations of traditional object detectors (Minderer et al., 2023; Zareian et al., 2021; Gu et al., 2021). This capability for real-time novelty recognition is essential for various real-world applications, including autonomous driving (Cao et al., 2023), biometric security (Bansal et al., 2021), and interactive assistants (Zou et al., 2023). A natural and early strategy for OVOD is to use CLIP-style cross-modal alignment to extract textual cues to recognize unseen categories (Radford et al., 2021).

The emergence of large language models (LLMs) has introduced new perspectives for OVOD by enabling detectors to benefit from richer and more expressive semantic supervision (Xu et al., 2023; Fu et al., 2025). Early attempts rely on prompt engineering to directly incorporate LLM-derived knowledge, but such designs require both the LLM and the detector to operate during inference, which significantly undermines efficiency. To mitigate this issue, knowledge distillation (KD) has gained increasing attention as a practical mechanism for transferring semantic knowledge from large vision-language models (LVLMs) into lightweight detectors, thereby maintaining open-vocabulary recognition capability while reducing inference cost. A representative example is ViLD (Gu et al., 2021), which employs a VLM to produce text embeddings of category names and distills their alignment with visual features into the detector. Following this paradigm, subsequent works such as DK DETR (Li et al., 2023) and DetCLIP (Yao et al., 2022) further improve visual semantic alignment

---

* Equal contribution.   † Corresponding authors.

to enhance detection of novel categories. Despite their effectiveness on certain benchmarks, existing distillation-based approaches remain closely coupled with textual encoders, which leaves unresolved trade-offs between inference latency and open-world generalization (Ma et al., 2025; Xiao et al., 2025). The limitation becomes clearer in complex scenarios and exposes two key challenges.

First, existing approaches often incur substantial computational overhead, as they rely on large text encoders or LLM-based prompt engineering at inference time to generate textual cues for novel categories, which limits their practical value (Liu et al., 2024b). Second, multimodal fusion designs inherently introduce a trade-off between closed-set detection accuracy and open-world generalization (Zareian et al., 2021; Gu et al., 2021). This trade-off originates from an optimization conflict: aggressively adapting features to seen categories tends to bias the model toward closed-set objectives, thereby weakening the cross-modal alignment required for recognizing unseen classes (Zhang et al., 2024; Fang et al., 2025). As a result, existing methods often improve one aspect at the expense of the other, motivating us to propose the Decoupled Cognition DETR (**DeCo-DETR**) framework.

To address the first challenge of computational bottleneck, we propose the **Dynamic Hierarchical Concept Pool** (DHCP). Instead of repeatedly invoking heavy text encoders at inference time, DHCP constructs a compact and reusable semantic prototype space that serves as a lightweight proxy for visual–language knowledge. Specifically, we leverage LLaVA (Liu et al., 2024a; 2023a;b) to obtain region-level semantic descriptions and project them into a shared embedding space via vision–language alignment, retaining only reliable cross-modal correspondences. Based on these aligned representations, we organize semantic concepts into a hierarchical prototype structure that captures both coarse category-level semantics and fine-grained variations. To further maintain robustness under distribution shifts, the prototype space is continuously refined during training through a momentum-based update mechanism. By decoupling semantic representation from online text encoding, DHCP enables efficient knowledge reuse and significantly reduces inference overhead.

To address the second challenge of task compromise between closed-set precision and open-world generalization, we introduce a decoupled cognition framework that disentangles representation learning from optimization dynamics. On the representation side, we design **Hierarchical Knowledge Distillation** (Hi-Know DPA), which aligns detector queries with the semantic prototype space through a learnable projection and structured aggregation, allowing each query to incorporate multi-granularity semantic cues while preserving its spatial sensitivity. On the optimization side, we propose **Parametric Decoupling Training** (PD-DuGi), which separates localization and semantic alignment into parallel training streams. By isolating gradient flows between the two objectives and adopting a progressive weighting strategy, the model mitigates interference during joint optimization. Together, these components operate in a coordinated manner, enabling DeCo-DETR to achieve a favorable balance between OVOD performance and inference efficiency.

The contributions can be summarized as follows:

- We reveal two critical flaws in existing open-vocabulary detection: 1) Heavy reliance on text encoders and LLM prompting causes high inference latency; 2) Multimodal fusion forces significant trade-offs between closed-set precision and open-world generalization.

- To address these issues, we propose DeCo-DETR. It eliminates text encoder dependency via the **Dynamic Hierarchical Concept Pool**, solving computational bottlenecks in multimodal fusion during inference time; and enhances generalization in open scenarios through **Hierarchical Knowledge Distillation** and **Parametric Decoupling Training**.

- We conduct extensive experiments on multiple open-vocabulary detection benchmarks, including OV-COCO and OV-LVIS. DeCo-DETR achieves competitive performance on all benchmarks, delivering significant improvements of 3.1 to 5.8 points in novel class APs while maintaining efficient 135ms inference. These results demonstrate DeCo-DETR's superior performance and generalization capabilities. Comprehensive ablation studies further validate the advantages of our novel design, providing valuable insights for the DETR-based detection paradigm and establishing a new foundation for future research.

## 2 RELATED WORK

**Open-vocabulary Object Detection (OVOD).** OVOD, formalized in (Zareian et al., 2021), uses image–caption data and base-class annotations to detect arbitrary categories, outperforming both

zero-shot and weakly supervised methods (Cai et al., 2022; Yao et al., 2021). Advances in vision–language models (VLMs) pre-trained on web-scale data (Radford et al., 2021; Jia et al., 2021) significantly improved OVOD. One approach leverages VLM knowledge to generate pseudo-labels for novel classes (Zhou et al., 2022; Liu et al., 2024b), using external sources or existing datasets like LVIS (Gupta et al., 2019), VL-PLM (Zhao et al., 2022). Another refines VLM interaction through learnable prompts (DetPro (Khattak et al., 2024), PromptDet (Feng et al., 2022)), surpassing static CLIP templates. However, these strategies incur high computational costs and incomplete knowledge transfer (Zhu & Chen, 2024). Thus, how to embed rich open-vocabulary semantics into lightweight detectors has become a focus (Rasheed et al., 2022; Ma et al., 2022; Gu et al., 2021).

**Visual Knowledge Distillation.** Knowledge distillation (KD) effectively transfers capabilities from large teacher models into compact student models (Xu et al., 2024), addressing the growing demand for efficient multimodal functionality on resource-constrained devices (Laroudie et al., 2023). For instance, TinyCLIP (Wu et al., 2023a) boosts open-vocabulary performance through advanced affinity mimicking and weight inheritance derived from CLIP. Subsequent research expands specialized KD strategies to lightweight detectors, enabling more practical deployment of VLMs while preserving their generalization capabilities (Pei et al., 2023; Li et al., 2024b). Recent studies also investigate robustness and generalization of distillation, showing that modality-gap stabilization, task-vector merging, and adversarial or subspace alignment can improve zero-shot robustness and accuracy–robustness trade-offs (Zhang et al., 2019; Dong et al., 2025a;d;b;c; Rade & Moosavi-Dezfooli, 2022). Robust distillation and representation learning further improve generalization under adversarial or data-scarce settings (Ilyas et al., 2019; Bai et al., 2021; Dong et al., 2024b;c).

**Knowledge Distillation for OVOD.** KD is highly effective beyond tasks like semantic segmentation (Ji et al., 2025) and visual reasoning (Aditya et al., 2019), showing significant impact in OVOD. ViLD (Gu et al., 2021) successfully distills a classification-based VLM into a two-stage detector, enhancing generalization. DK-DETR (Li et al., 2023) further improves OVOD precision by distilling VLM knowledge into DETR which is a transformer-based architecture specifically designed for object detection (Carion et al., 2020). KD has thus become mainstream in OVOD (Wang et al., 2023b; Wu et al., 2023b; Rasheed et al., 2022). However, reliance on textual cues from large models limits generalization and efficiency. CAKE (Ma et al., 2025) mitigates textual dependence but struggles with fine-grained detection. DeCo-DETR addresses these gaps by implementing a vision-centric mechanism that enhances perception without heavy textual dependency.

## 3 METHOD

### 3.1 PROBLEM DEFINITION

Open-vocabulary Object Detection (OVOD) aims to detect and recognize objects from both seen and unseen categories. Formally, the training set is defined as:

$$\mathcal{T} = \{(I_i, g_i)\}_{i=1}^N, \tag{1}$$

where $I_i \in \mathbb{R}^{H \times W \times 3}$ denotes an input image, and $g_i = \{(b_{i,k}, c_{i,k})\}_{k=1}^{K_i}$ represents the ground-truth annotations, where $b_{i,k} \in \mathbb{R}^4$ is the bounding box and $c_{i,k} \in C_{\text{base}}$ is the category label belonging to the set of known categories $C_{\text{base}}$. During inference, the detector is required to predict:

$$\{(b_j, c_j)\}_{j=1}^M, \quad c_j \in C_{\text{base}} \cup C_{\text{novel}}, \tag{2}$$

where $C_{\text{novel}}$ denotes unseen categories and $C_{\text{base}} \cap C_{\text{novel}} = \emptyset$. The core challenge of OVOD lies in enabling the detector to generalize beyond training labels and recognize novel semantic concepts through cross-modal knowledge transfer.

### 3.2 FRAMEWORK OVERVIEW

Existing OVOD approaches suffer from high computational overhead and inherent task conflicts between localization and semantic alignment. To address these limitations, we propose **DeCo-DETR**, a decoupled cognition framework that efficiently transfers open-set knowledge from large vision-language models (LVLMs) into a compact detector without requiring text encoders at inference time. As illustrated in Figure 1, DeCo-DETR consists of three key components:

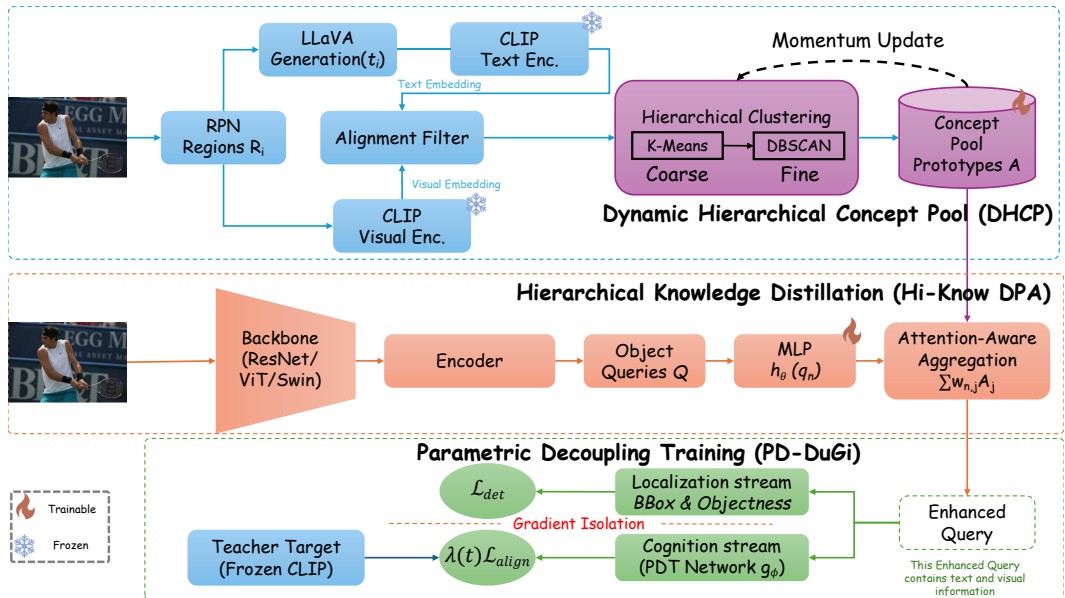

Figure 1: Three-stage pipeline of DeCo-DETR. (a) DHCP constructs a hierarchical prototype memory from region-level descriptions via LLaVA generation and CLIP-based filtering, capturing both coarse and fine-grained semantics. (b) Hi-Know DPA projects detector queries into the shared embedding space and enhances them through prototype aggregation for efficient open-set knowledge transfer. (c) PD-DuGi decouples localization and semantic alignment into two optimization streams, mitigating task interference. At inference, the prototype pool provides semantic priors, and the decoupled decoder jointly predicts bounding boxes and category semantics without a text encoder.

**Dynamic Hierarchical Concept Pool** (DHCP) constructs a self-evolving set of semantic prototypes by distilling LLaVA-generated region descriptions filtered via CLIP alignment, replacing costly text encoders. **Hierarchical Knowledge Distillation** (Hi-Know DPA) aligns visual features with hierarchical prototypes through a projection-based decoupling mechanism, enabling fine-grained semantic grounding. **Parametric Decoupling Training** (PD-DuGi) resolves optimization conflicts via dual-stream gradient isolation, allowing localization and semantic alignment to be learned independently yet synergistically. At inference, semantic knowledge is provided by the learned prototype pool, while a dual-stream decoder performs localization and semantic reasoning in parallel. For clarity, the algorithmic implementations of these modules are presented as pseudocode in the Appendix A.3.

## 3.3 DYNAMIC HIERARCHICAL CONCEPT POOL

To construct a compact yet expressive semantic memory for open-vocabulary detection, we propose a **Dynamic Hierarchical Concept Pool** (DHCP), which progressively builds and refines a set of vision-language prototypes. The process consists of two stages: (1) an offline initialization stage that establishes a reliable cross-modal semantic space and organizes it into hierarchical prototypes, and (2) an online update stage that continuously adapts the prototype representations during training.

**Stage I: Initialization via cross-modal alignment and hierarchical distillation** We first construct a high-quality semantic embedding set through cross-modal alignment. Given a training dataset $\mathcal{D}$, for each image $I \in \mathcal{D}$, we extract region proposals $\{R_i\}_{i=1}^N$, where $R_i \in \mathbb{R}^{h_i \times w_i \times 3}$ denotes the $i$-th image region. For each region, we generate a free-form textual description:

$$t_i = \text{LLaVA}(R_i), \tag{3}$$

where $t_i$ denotes the generated text. To bridge the modality gap, both image regions and text descriptions are projected into a shared embedding space using CLIP:

$$v_i = f_{\text{CLIP}}^{\text{img}}(R_i), \quad u_i = f_{\text{CLIP}}^{\text{txt}}(t_i), \tag{4}$$

where $v_i \in \mathbb{R}^d$ and $u_i \in \mathbb{R}^d$ denote image and text embeddings, respectively, and $d$ is the embedding dimension. To ensure semantic consistency, we retain only high-confidence text embeddings:

$$\mathcal{T} = \{u_i \mid \cos(v_i, u_i) > \delta\}, \tag{5}$$

where $\delta$ is a similarity threshold and $\cos(\cdot, \cdot)$ denotes cosine similarity. The resulting set $\mathcal{T}$ forms a filtered semantic embedding pool. Based on $\mathcal{T}$, we construct hierarchical prototypes to capture multi-level semantics. First, we apply K-Means clustering to obtain coarse-grained clusters:

$$C_{\text{coarse}} = \text{K-Means}(\mathcal{T}, k = M_1), \tag{6}$$

where $M_1$ denotes the number of coarse prototypes. The centroids of these clusters form the initial coarse-level prototypes. Then, for each cluster $c \in C_{\text{coarse}}$, we apply local density-based clustering:

$$C_{\text{fine}} = \text{DBSCAN}(c), \tag{7}$$

where $C_{\text{fine}}$ denotes fine-grained sub-clusters within $c$. The centroids of these sub-clusters capture fine-grained semantic variations. The final prototype matrix is constructed as: $A \in \mathbb{R}^{d \times M}$, where $M = M_1 + M_2$ denotes the total number of prototypes, and each column $A_j \in \mathbb{R}^d$ represents a semantic prototype. This hierarchical design enables the representation of both global category-level concepts and local attribute-level details.

**Stage II: Online Prototype Update** To adapt the prototype space to evolving training distributions, we introduce a dynamic update mechanism. During training, given a batch of aligned embeddings $\{e_i\}_{i=1}^K$, where $e_i \in \mathbb{R}^d$, we compute their soft assignment to the prototype set:

$$D_{i,j} = \frac{\exp(\tau^{-1} \cos(e_i, A_j))}{\sum_{k=1}^M \exp(\tau^{-1} \cos(e_i, A_k))}, \tag{8}$$

where $D_{i,j}$ is the assignment weight of embedding $e_i$ to prototype $A_j$, and $\tau$ is the temperature controlling distribution sharpness. Each prototype is updated using a momentum-based rule:

$$A_j \leftarrow \gamma A_j + (1 - \gamma)\text{LayerNorm}\left(\sum_{i=1}^K D_{i,j} e_i\right), \tag{9}$$

where $\gamma \in [0, 1]$ controls the update rate, and LayerNorm ensures numerical stability.

This online refinement allows the concept pool to continuously incorporate new semantic patterns while preserving previously learned structures, resulting in a stable and adaptive semantic memory for open-vocabulary detection.

### 3.4 HIERARCHICAL KNOWLEDGE DISTILLATION

The above DHCP provides a structured semantic prototype space that encodes multi-granularity concepts. Building upon this semantic memory, we introduce **Hierarchical Knowledge Distillation** (Hi-Know DPA) to bridge detector features and the prototype space, enabling efficient transfer of open-vocabulary knowledge from pretrained VLMs.

Given an input image $I$, the backbone extracts feature maps: $\Phi(I) \in \mathbb{R}^{H \times W \times C}$, where $H, W$ denote spatial resolution and $C$ denotes the channel dimension. A transformer decoder then produces a set of object queries $\mathcal{Q} = \{q_n\}_{n=1}^N, q_n \in \mathbb{R}^C$, where $N$ denotes the number of queries and each $q_n$ encodes a candidate object representation.

To align these visual queries with the semantic prototypes, we use a learnable projection network $h_\theta : \mathbb{R}^C \to \mathbb{R}^d$, which maps each query into the shared embedding space:

$$\hat{q}_n = h_\theta(q_n), \tag{10}$$

where $\hat{q}_n \in \mathbb{R}^d$ denotes the projected query and $d$ is the embedding dimension consistent with the prototype space $A \in \mathbb{R}^{d \times M}$. This projection establishes a unified representation space where visual features and semantic prototypes become directly comparable. Given the projected queries, we compute their similarity with all prototypes to obtain a semantic assignment distribution:

$$w_{n,j} = \frac{\exp(\alpha^{-1} \cos(\hat{q}_n, A_j))}{\sum_{k=1}^M \exp(\alpha^{-1} \cos(\hat{q}_n, A_k))}, \tag{11}$$

where $w_{n,j}$ denotes the assignment weight between query $\hat{q}_n$ and prototype $A_j$, $\alpha$ is a temperature parameter controlling distribution sharpness, and $\cos(\cdot, \cdot)$ denotes cosine similarity.

Based on these weights, we aggregate semantic information from the prototype space to form enhanced query representations:

$$r_n = \sum_{j=1}^{M} w_{n,j} A_j + \text{MLP}(\hat{q}_n), \tag{12}$$

where $r_n \in \mathbb{R}^d$ denotes the semantic-enhanced query, and the MLP term preserves the original visual information via a residual connection. This design allows each query to dynamically integrate both coarse-grained and fine-grained semantic cues from the hierarchical prototype space.

To further guide this alignment, we utilize teacher-student distillation based on a frozen CLIP. Let $z_n^{\text{CLIP}} \in \mathbb{R}^d$ denote the CLIP visual embedding associated with the $n$-th query region. Given the textual prototype matrix $P = \{p_j\}_{j=1}^{M}$, where each $p_j \in \mathbb{R}^d$ is derived from category names and LLaVA-generated descriptions, the teacher produces a teacher-guided semantic distribution:

$$\tilde{w}_n = \text{Softmax}(\tau^{-1} \cos(z_n^{\text{CLIP}}, P)), \tag{13}$$

where $\tau$ is a temperature parameter. For Hi-Know DPA, the overall training objective is:

$$\mathcal{L}_{DPA} = \mathcal{L}_{\text{det}} + \lambda_{\text{KL}} \sum_{n=1}^{N} \text{KL}(\tilde{w}_n \| w_n) + \lambda_{\text{align}} \mathcal{L}_{\text{align}}, \tag{14}$$

where $\mathcal{L}_{\text{det}}$ denotes the standard DETR detection loss, $w_n \in \mathbb{R}^M$ denotes the predicted prototype distribution, $\text{KL}(\cdot \| \cdot)$ denotes KL divergence, and $\mathcal{L}_{\text{align}}$ denotes an auxiliary alignment loss for stabilizing feature matching.

Overall, Hi-Know DPA establishes an interaction between detector queries and semantic prototypes, enabling effective knowledge transfer while preserving the visual capability of the detector.

### 3.5 Parametric Decoupling Training

While hierarchical distillation enables effective semantic grounding, a fundamental challenge remains: the optimization objectives of object localization and open-vocabulary alignment are inherently different and often conflicting. Localization requires precise spatial discrimination, whereas semantic alignment emphasizes cross-modal generalization. Directly optimizing both objectives within a shared parameter space can lead to representation interference and degraded performance.

To address this issue, we propose **Parametric Decoupling Training** (PD-DuGi), which separates the optimization of detection and semantic cognition through a dual-stream architecture with gradient isolation. Given the semantic-enhanced query features $\{r_n\}_{n=1}^{N}$, where $r_n \in \mathbb{R}^d$ is obtained from hierarchical prototype aggregation, we introduce a parametric semantic predictor:

$$g_\phi : \mathbb{R}^d \to \mathbb{R}^{|C_{\text{base}} \cup C_{\text{novel}}|}, \tag{15}$$

which maps each query into an open-vocabulary category space. It is then used to obtain the predicted category probability distribution:

$$t_n = \text{Softmax}(g_\phi(r_n)) \in \mathbb{R}^{|C_{\text{base}} \cup C_{\text{novel}}|}. \tag{16}$$

$g_\phi$ is implemented using multi-layer cross-attention blocks to capture correlations between prototypes and category embeddings. To disentangle the learning dynamics, we decouple the training process into two parallel streams operating on shared query representations but optimized separately.

**Semantic Alignment Stream** To prevent interference from detection optimization, we stop gradients from the alignment objective:

$$q'_n = \text{StopGradient}(q_n), \tag{17}$$

where $q'_n \in \mathbb{R}^C$ is treated as a constant with respect to alignment gradients.

The queries are then projected and semantically enriched:

$$\hat{q}_n = h_\theta(q'_n), \quad r_n = \text{PrototypeAggregation}(\hat{q}_n, A), \tag{18}$$

where $\hat{q}_n \in \mathbb{R}^d$ is the projected feature and $A \in \mathbb{R}^{d \times M}$ is the prototype matrix. $r_n$ then produces $t_n$ as described above via $g_\phi$. We utilize a frozen CLIP teacher to generate target semantic distributions to provide supervision:

$$T_{\text{teacher}} = \text{CLIP}_{\text{teacher}}(I, \text{Prompts}) \in \mathbb{R}^{|C_{\text{base}} \cup C_{\text{novel}}|}, \tag{19}$$

which encodes cross-modal similarity between the image and category prompts.

The alignment loss is defined as:

$$\mathcal{L}_{\text{align}} = \text{CrossEntropy}(t_n, T_{\text{teacher}}), \tag{20}$$

which updates only the semantic modules, including $g_\phi$ and $h_\theta$.

During PD-DuGi, we explicitly isolate the gradient flows of two streams: Gradients from $\mathcal{L}_{\text{det}}$ are restricted to the detection backbone and decoder, while gradients from $\mathcal{L}_{\text{align}}$ are restricted to the semantic projection and predictor. This separation prevents destructive interference between spatial localization and semantic alignment, effectively decoupling the visual manifold $\mathcal{V}$ and semantic manifold $\mathcal{S}$ while enabling their joint mapping to the output space $\mathcal{Y}$.

The overall training objective of PD-DuGi combines both streams:

$$\mathcal{L}_{PD} = \mathcal{L}_{\text{det}} + \lambda_{\text{align}}(t)\mathcal{L}_{\text{align}}, \tag{21}$$

where $\lambda_{\text{align}}(t)$ is a time-dependent weighting factor following a cosine annealing schedule. This curriculum strategy prioritizes stable detection learning in early stages and gradually emphasizes semantic alignment. At inference time, the model performs single-pass prediction without requiring text encoders. The learned prototype space and semantic predictor jointly enable open-vocabulary recognition while maintaining efficient detection.

## 4 EXPERIMENT

### 4.1 DATASETS AND EVALUATION METRICS

Following standard protocols in the OVOD literature (Jin et al., 2024; Zhou et al., 2022; Ma et al., 2025), we evaluate the effectiveness of DeCo-DETR on two widely adopted benchmarks: OV-COCO (Bansal et al., 2018) and OV-LVIS (Gu et al., 2021). These benchmarks are open-vocabulary variants derived from the popular MSCOCO (Lin et al., 2015) and LVIS datasets, respectively. OV-COCO contains 118,000 images, with 48 categories designated as base classes and 17 held out as novel classes for zero-shot evaluation. OV-LVIS uses the same images with LVIS annotations, where 866 frequent and common categories form the base set and 337 rare categories are treated as novel. This long-tail distribution better reflects real-world scenarios and poses a more challenging setting for OVOD. For OV-COCO, we report $AP^{50}$novel as the primary metric, along with $AP^{50}$base and overall $AP^{50}$. For OV-LVIS, we report $AP_r$, $AP_c$, and $AP_f$ for rare, common, and frequent categories, respectively, as well as the overall $AP$, all computed using standard box-based mAP. Details on V-OVD, G-OVD, C-OVD, and WS-OVD are provided in Appendix A.4.

### 4.2 IMPLEMENTATION DETAILS

We implement DeCo-DETR based on the standard DETR framework with ResNet-50, ViT-B/16, and Swin-T backbones. For DHCP, we construct a hierarchical prototype pool with $M_1 = 1203$ coarse-grained and $M_2 = 4800$ fine-grained prototypes ($M = 6003$ in total). The prototype matrix $A$ is updated online using the momentum coefficient $\gamma = 0.99$. The temperature parameter $\tau = 0.07$ is used in prototype assignment to control the sharpness of similarity distributions. For Hi-Know DPA, the projection network maps decoder queries into the shared embedding space of dimension $d$. The prototype assignment employs a temperature parameter $\alpha$. The weighting coefficients $\lambda_{\text{KL}}$ and $\lambda_{\text{align}}$ follow cosine annealing schedules, gradually shifting the training focus from detection to semantic alignment. For PD-DuGi, we adopt the dual-stream training strategy with cosine-annealed weighting $\lambda_{\text{align}}(t)$ to balance detection and alignment objectives over training.

Table 1: OV-COCO comparison ($AP_{50}$) across a wide range of widely used OVOD methods.

| Benchmark | Method | $\mathbf{AP}_{50}^{novel}$ | $\mathbf{AP}_{50}^{base}$ | $\mathbf{AP}_{50}$ |
|---|---|---|---|---|
| **V-OVD** | ViLD (Gu et al., 2021) | 29.4 | 52.6 | 48.9 |
| | OADP (Wang et al., 2023b) | 30.0 | 53.3 | 47.2 |
| | DK-DETR (Li et al., 2023) | 32.3 | **61.1** | **53.6** |
| | BARON (Wu et al., 2023b) | 33.1 | 54.8 | 49.1 |
| | LBP (Li et al., 2024a) | 37.8 | 58.7 | 53.2 |
| | OC-OVD (Rasheed et al., 2022) | 36.6 | 54.0 | 49.4 |
| | GOAT (Wang et al., 2023a) | 36.4 | 53.0 | 48.6 |
| | CAKE (Ma et al., 2025) | 38.2 | 58.0 | 52.8 |
| | **DeCo-DETR (Ours)** | **41.3** | 56.7 | 53.1 |
| **G-OVD** | OV-DETR (Zang et al., 2022) | 29.4 | **61.0** | 52.7 |
| | VL-PLM (Zhao et al., 2022) | 32.3 | 54.0 | 48.3 |
| | OADP (Wang et al., 2023b) | 35.6 | 55.8 | 50.5 |
| | LP-OVOD (Pham, 2024) | 40.5 | 60.5 | **55.2** |
| | CLIM (Wu et al., 2024) | 25.7 | 42.5 | - |
| | CCKT-Det(Zhang et al., 2025) | - | - | 53.2 |
| | RALF (Kim et al., 2024) | 41.3 | 54.3 | 50.9 |
| | CAKE (Ma et al., 2025) | 39.1 | 58.1 | 53.1 |
| | **DeCo-DETR (Ours)** | **47.1** | 60.2 | 55.0 |
| **C-OVD** | RegionCLIP (Zhong et al., 2022) | 26.8 | 54.8 | 47.5 |
| | CoDet (Ma et al., 2023) | 30.6 | 52.3 | 46.6 |
| | BARON (Wu et al., 2023b) | 35.8 | 58.2 | 52.3 |
| | BIRDet (Zeng et al., 2024) | **46.2** | **63.0** | **58.6** |
| | CAKE (Ma et al., 2025) | 41.3 | 60.2 | 55.3 |
| | **DeCo-DETR (Ours)** | 44.9 | 59.8 | 56.3 |
| **WS-OVD** | Detic (Zhou et al., 2022) | 28.4 | 53.8 | 47.2 |
| | GOAT (Wang et al., 2023a) | 36.4 | 53.0 | 48.6 |
| | OC-OVD (Rasheed et al., 2022) | 36.6 | 54.0 | 49.4 |
| | CAKE (Ma et al., 2025) | 41.8 | **60.6** | 55.7 |
| | **DeCo-DETR (Ours)** | **45.5** | 60.5 | **57.1** |

Table 2: OV-LVIS comparison ($AP$) across a wide range of widely used OVOD methods.

| Method | $\mathbf{AP_r}$ | $\mathbf{AP_c}$ | $\mathbf{AP_f}$ | $\mathbf{AP}$ |
|---|---|---|---|---|
| DetPro (Du et al., 2022) | 20.8 | 27.8 | 32.4 | 28.4 |
| VLDet (Lin et al., 2022) | 21.7 | 29.8 | 34.3 | 30.1 |
| OC-OVD (Rasheed et al., 2022) | 21.1 | 25.0 | 29.1 | 25.9 |
| OADP (Wang et al., 2023b) | 21.9 | 28.4 | 32.0 | 28.7 |
| CORA (Wu et al., 2023c) | 22.2 | 32.0 | **40.2** | 33.5 |
| BARON (Wu et al., 2023b) | 23.2 | 29.3 | 32.5 | 29.5 |
| CoDet (Ma et al., 2023) | 23.4 | 30.0 | 34.6 | 30.7 |
| LBP (Li et al., 2024a) | 24.1 | 29.5 | 32.8 | 29.9 |
| LP-OVOD (Pham, 2024) | 19.3 | 26.1 | 29.4 | 26.2 |
| Mamba (Wang et al., 2025) | 29.3 | **34.2** | 36.8 | 35.0 |
| BIRDet (Zeng et al., 2024) | 26.0 | 21.7 | 29.5 | 25.5 |
| RALF (Kim et al., 2024) | 21.9 | 26.2 | 29.1 | 26.6 |
| **DeCo-DETR (Ours)** | **29.4** | 33.1 | 38.9 | **35.2** |

The decoder consists of 6 Transformer layers with 8 attention heads each. We train the model with a total batch size of 64 (8 samples per GPU across 8 × NVIDIA A100 GPUs). Due to hardware constraints, we apply dataset cropping in certain ablation experiments. During inference, all experiments are conducted on a single NVIDIA RTX 4090 GPU.

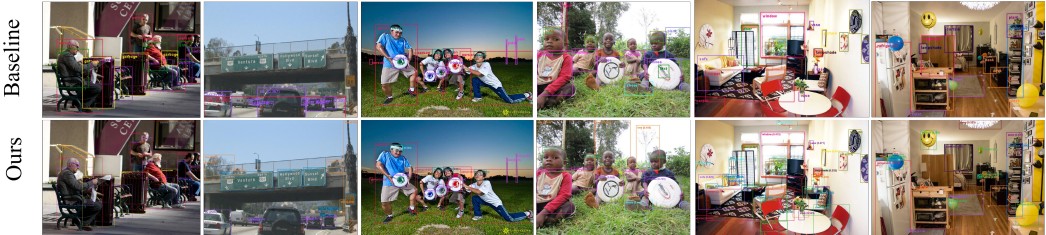

Figure 2: Qualitative comparison between DeCo-DETR and the baseline. DeCo-DETR shows stronger open-vocabulary generalization by accurately localizing and recognizing novel categories.

Table 3: Inference latency, GFLOPs, and parameter size across three backbone architectures.

| Method | Latency (ms/img) | | | GFLOPs | | | Params (M) | | |
|---|---|---|---|---|---|---|---|---|---|
| | R50 | ViT | Swin | R50 | ViT | Swin | R50 | ViT | Swin |
| Deformable DETR (Zhu et al., 2020b) | 120 | 210 | 220 | 220 | 320 | 325 | 41 | 87 | 95 |
| DetPro (Du et al., 2022) | 140 | 250 | 260 | 240 | 340 | 345 | 45 | 91 | 100 |
| UP-DETR (Dai et al., 2021) | 115 | 205 | 215 | 215 | 315 | 320 | 40 | 85 | 92 |
| **DeCo-DETR (Ours)** | **135** | **240** | **250** | **235** | **335** | **340** | **44** | **90** | **97** |

## 4.3 MAIN RESULTS

DeCo-DETR achieves strong zero-shot OVOD performance on both OV-COCO and OV-LVIS, establishing it as an effective OVOD method. As shown in Table 1, DeCo-DETR attains **41.3%** $AP_{50}^{novel}$ on OV-COCO, surpassing the strongest baseline LBP (37.8%) by **+3.5 points**, while the overall $AP_{50}$ outperforms all baselines. On the challenging long-tailed OV-LVIS dataset (Table 2), DeCo-DETR achieves **29.4%** $AP_r$ for rare classes, and sets a new record with an overall AP of **35.2%**. These results demonstrate DeCo-DETR's capability to mitigate classification bias in long-tailed distributions while maintaining high accuracy for common and frequent classes. Moreover, it balances accuracy and efficiency. With ResNet-50 backbone (Table 3), inference latency increases by only **5-15ms**, computation (GFLOPs) by **6.8%**, and parameters by **7.3%** (44M vs. 41M). Compared to ViLD (140ms/img) and DetPro (250ms/img), DeCo-DETR achieves a better balance between accuracy and efficiency. We provide several qualitative examples of DeCo-DETR in Figure 2.

## 4.4 ABLATION STUDY

In this section, we conduct a series of ablation studies to evaluate the contribution of each component, with the main results shown in Table 4 and additional results provided in Appendix A.6.

**DHCP.** Incorporating multigranular prototypes (1,203 coarse + 4,800 fine) improves $AP_{50}^{novel}$ by **2.5%** compared to using a single-level prototype pool. This result underscores the effectiveness of hierarchical semantic abstraction: coarse-level prototypes capture broad inter-class distinctions, while fine-level prototypes model subtle intra-class variations. By jointly leveraging these multiscale semantic features, the model is better positioned to generalize to novel categories under limited supervision, leading to a notable performance gain.

**PD-DuGi.** The integration of the PD-DuGi mechanism yields a comprehensive improvement across all metrics, validating the necessity of resolving task conflicts in open-vocabulary detection. The introduction of dual-stream gradient isolation boosts $AP_{50}^{novel}$ from 36.6% to 37.5% (+0.9%) and, notably, increases $AP_{50}^{base}$ from 54.0% to 55.1% (+1.1%). This simultaneous gain suggests that sharing a unified feature space for both localization and semantic alignment often leads to optimization interference, where the gradients for semantic adaptation may degrade the spatial features required for precise bounding box regression. PD-DuGi effectively mitigates this issue by explicitly isolating the optimization paths; it allows the cognition branch to learn robust semantic representations for novel categories without distorting the structural features essential for base category localization, thereby achieving a superior trade-off between open-world generalization and closed-set precision.

Table 4: Main ablation studies.

| Configuration | $AP_{50}^{novel}$ | $AP_{50}^{novel}$ | $AP_{50}$ |
|---|---|---|---|
| 1. Baseline only | 30.4 | 52.6 | 46.8 |
| 2. + Hierarchical DHCP | 36.6 | 54.0 | 49.4 |
| 3. + PD-DuGi (Gradient Isolation) | 37.5 | 55.1 | 50.5 |
| 4. + Cosine $\lambda(t)$ (Full Model) | 41.3 | 55.5 | 51.0 |

**Cosine Annealing Weights.** Dynamically balancing detection and alignment losses using a cosine annealing schedule improves $AP_{50}$ by an additional **1.6%**. The time-dependent coefficient $\lambda(t)$ initially emphasizes the alignment loss to encourage robust feature embedding early in training, and gradually shifts focus toward the detection loss to refine localization and classification boundaries. This smooth transition alleviates potential conflicts between the two objectives, promoting more stable convergence and improved detection accuracy on novel categories.

**Impact of Different LVLMs**: We further investigate the impact of varying scales of vision-language models (VLMs) on detection performance (see Table 6). Experimental results indicate that when using smaller models (e.g., LLaVA-1.5 7B), there is a noticeable limitation on the detection performance for novel classes ($AP_{50}^{novel}$), which is only 30.1%. However, when the model scale increases to 13B or larger (e.g., LLaVA-1.5 13B, LLaVA-NEXT 13B, Qwen2.5-VL 32B), $AP_{50}^{novel}$ stabilizes between 38.2%–38.9%, showing significant improvement over the 7B model. This suggests that once the model parameter count exceeds a certain threshold (around 13B), further increases in parameters have a negligible impact on detection accuracy. Therefore, in practical deployment, a moderately sized VLM can be selected to balance performance and computational cost.

**Impact of Queries and Prototypes.** Table 7 investigates the impact of decoder query quantity ($N$) and prototype granularity ($M_2$). Regarding the number of queries, increasing $N$ from 300 to 2000 yields a substantial performance gain of $+4.8\ AP_{novel}$. Thanks to the parallel nature of the Transformer decoder, this improvement incurs only a marginal latency overhead ($\sim$10ms). Notably, even with a reduced set of $N = 300$, our method achieves $36.5\%\ AP_{novel}$, significantly outperforming previous state-of-the-art methods like ViLD (29.4%). Regarding prototype scale, the fine-grained units ($M_2$) prove critical for open-vocabulary generalization. Removing them ($M_2 = 0$) causes a sharp drop of $10.5$ points in $AP_{novel}$, validating the effectiveness of DHCP. Conversely, doubling the fine-grained units to 9600 yields diminishing returns ($+0.2\%\ AP_{novel}$) while increasing memory usage and latency, confirming that $M_2 = 4800$ is the optimal configuration.

**Efficiency Analysis.** Table 8 presents a comprehensive comparison of inference latency and detection performance. Compared to fusion-based methods like Grounding DINO, which rely on computationally heavy text encoders (e.g., BERT-Base) and suffer from high latency ($\sim$280ms), our DeCo-DETR eliminates the text encoder dependency during inference. This architectural advantage results in a significant speedup of approximately $2\times$ (135ms vs. 280ms) while maintaining competitive accuracy (41.3% vs. 42.1% $AP_{novel}$). Furthermore, among distillation-based and decoupled methods, DeCo-DETR has strong performance, by increasing $AP_{novel}$ to 41.3 while reducing latency to 135ms (7.4 FPS). These results demonstrate that DeCo-DETR establishes a superior efficiency-accuracy trade-off, making it highly suitable for real-time open-vocabulary applications.

## 5 CONCLUSION

In this work, we present **DeCo-DETR**, a novel framework for open-vocabulary object detection that integrates semantic knowledge into a compact detection pipeline. By leveraging a structured semantic representation and a decoupled training paradigm, our approach enables effective knowledge transfer from pre-trained multimodal models while maintaining strong detection capability. Extensive experiments on LVIS and COCO demonstrate competitive zero-shot detection performance, together with improved efficiency by removing the need for auxiliary text encoders at inference. Beyond these results, DeCo-DETR provides a simple and scalable design for incorporating semantic understanding into detection models, offering a promising direction for open-world perception.

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

# A    APPENDIX

## A.1    ETHICS STATEMENT

This work adheres to the ICLR Code of Ethics. Our study does not involve human subjects, personal or sensitive data. All datasets used in this paper (e.g., COCO, LVIS) are publicly available and widely adopted in the research community, and we strictly follow their licenses and intended usage. The proposed DeCo-DETR framework is designed for academic exploration of open-vocabulary object detection. Potential misuse of the model in safety-critical or surveillance scenarios is outside the scope of this research, and we strongly encourage responsible and ethical use in line with research integrity principles.

## A.2    REPRODUCIBILITY STATEMENT

We make every effort to ensure the reproducibility of our results. Full training details, including model architectures, hyperparameters, and optimization schedules, are provided in the main paper and appendix. The experimental settings cover key modules such as Dynamic Hierarchical Concept Pool construction, Hierarchical Knowledge Distillation, and Parametric Decoupling Training, with clear descriptions of dataset preprocessing and evaluation protocols. Our implementation is based on PyTorch and standard detection frameworks. To facilitate replication, we will release the source code, configuration files, and pre-trained models upon publication. All reported results can be reproduced using the provided settings and supplementary material.

## A.3    METHODOLOGY DETAILS

The pseudocode for the algorithms implementing the Dynamic Hierarchical Concept Pool, Hierarchical Knowledge Distillation, and Parametric Decoupling Training is presented below:

---

**Algorithm 1** Dynamic Hierarchical Concept Pool (DHCP)

---

**Require:** Training images $\mathcal{D}$, Backbone, LLaVA, CLIP (frozen)
**Require:** Hyperparameters: threshold $\delta$, momentum $\gamma$, temperature $\tau$
**Ensure:** Prototype matrix $A \in \mathbb{R}^{d \times M}$

1: **Stage I: Offline Initialization**
2: Initialize $\mathcal{T} \leftarrow \emptyset$
3: **for** each image $I \in \mathcal{D}$ **do**
4:     Extract regions $\{R_i\}$
5:     $t_i \leftarrow \text{LLaVA}(R_i)$
6:     $v_i \leftarrow f_{\text{CLIP}}^{\text{img}}(R_i),\ u_i \leftarrow f_{\text{CLIP}}^{\text{txt}}(t_i)$
7:     **if** $\cos(v_i, u_i) > \delta$ **then**
8:         $\mathcal{T} \leftarrow \mathcal{T} \cup \{u_i\}$
9:     **end if**
10: **end for**
11: $C_{\text{coarse}} \leftarrow \text{K-Means}(\mathcal{T}, k = M_1)$
12: $A \leftarrow \text{Centroids}(C_{\text{coarse}})$
13: **for** each cluster $c \in C_{\text{coarse}}$ **do**
14:     $C_{\text{fine}} \leftarrow \text{DBSCAN}(c)$
15:     Append centroids of $C_{\text{fine}}$ to $A$
16: **end for**
17: **Stage II: Online Update**
18: **while** training **do**
19:     Receive embeddings $\{e_i\}$
20:     $D_{i,j} \leftarrow \dfrac{\exp(\tau^{-1}\cos(e_i, A_j))}{\sum_k \exp(\tau^{-1}\cos(e_i, A_k))}$
21:     **for** each prototype $A_j$ **do**
22:         $A_j \leftarrow \gamma A_j + (1 - \gamma)\text{LayerNorm}\left(\sum_i D_{i,j}e_i\right)$
23:     **end for**
24: **end while**

---

---

**Algorithm 2** Hierarchical Knowledge Distillation (Hi-Know DPA)

---

**Require:** Training set $\mathcal{D}$, image $I$
**Require:** Student: Backbone, projection $h_\theta$, prototypes $A \in \mathbb{R}^{d \times M}$
**Require:** Teacher: CLIP (frozen), text prototypes $P \in \mathbb{R}^{M \times d}$
 1: **while** not converged **do**
 2:    Sample batch $\mathcal{B}$
 3:    **for** each $I \in \mathcal{B}$ **do**
 4:        $\Phi(I) \leftarrow \text{Backbone}(I)$
 5:        $\mathcal{Q} \leftarrow \text{Decoder}(\Phi(I))$
 6:        $\hat{q}_n \leftarrow h_\theta(\mathcal{Q})$
 7:        $w_n \leftarrow \text{Softmax}(\alpha^{-1} \cos(\hat{q}_n, A))$
 8:        $r_n \leftarrow \sum_j w_{n,j} A_j + \text{MLP}(\hat{q}_n)$
 9:        $\tilde{w}_n \leftarrow \text{Softmax}(\tau^{-1} \cos(\hat{q}_n, P))$
10:        $\mathcal{L} \leftarrow \mathcal{L}_{\text{det}} + \lambda_{\text{KL}} \sum_n \text{KL}(w_n \| \tilde{w}_n) + \lambda_{\text{align}} \mathcal{L}_{\text{align}}$
11:    **end for**
12:    Update parameters
13: **end while**

---

**Algorithm 3** Parametric Decoupling Training (PD-DuGi)

---

**Require:** Image $I$, Ground Truth $Y$, Student Model (Backbone, Decoder, PDT), Teacher CLIP
**Require:** Scheduler $\lambda_{align}(t)$
**Ensure:** Optimized parameters $\theta$
 1: **Forward**
 2: $\Phi(I) \leftarrow \text{Backbone}(I)$
 3: $\mathcal{Q} = \{q_n\} \leftarrow \text{Decoder}(\Phi(I))$
 4: **Localization Stream**
 5: $Y_{\text{pred}} \leftarrow \text{DetectionHead}(q_n)$
 6: $\mathcal{L}_{det} \leftarrow \text{Loss}_{\text{Hungarian}}(Y_{\text{pred}}, Y)$
 7: **Semantic Stream**
 8: $q'_n \leftarrow \text{StopGradient}(q_n)$
 9: $\hat{q}_n \leftarrow h_\theta(q'_n)$
10: $r_n \leftarrow \text{PrototypeAggregation}(\hat{q}_n, A)$
11: $t_n \leftarrow \text{Softmax}(g_\phi(r_n))$
12: $T_{\text{teacher}} \leftarrow \text{CLIP}(I, \text{Prompts})$
13: $\mathcal{L}_{align} \leftarrow \text{CrossEntropy}(t_n, T_{\text{teacher}})$
14: **Optimization**
15: $\lambda \leftarrow \lambda_{align}(t)$
16: $\mathcal{L}_{total} \leftarrow \mathcal{L}_{det} + \lambda \mathcal{L}_{align}$
17: Update $\theta$

---

**Localization Stream** The original decoder queries $\{q_n\}$ are directly used for object detection. The detection head predicts bounding boxes and objectness scores:

$$Y_{\text{pred}} = \text{DetectionHead}(q_n), \tag{22}$$

where $Y_{\text{pred}}$ denotes predicted object instances. The detection loss is defined as:

$$\mathcal{L}_{\text{det}} = \text{Loss}_{\text{Hungarian}}(Y_{\text{pred}}, Y), \tag{23}$$

where $Y$ denotes ground-truth annotations. This loss updates only the backbone and decoder parameters, preserving spatial modeling capability. The student prototypes $A$ and teacher prototypes $P$ share the same index dimension $M$ because they are derived from the same set of multi-modal clusters in the joint CLIP embedding space. This correspondence is established as follows:

1. **Joint Clustering:** We perform clustering on the aligned pairs of region visual features and text embeddings (filtered by CLIP). This partitions the data into $M$ clusters, where each cluster $j \in \{1, \dots, M\}$ represents a specific shared semantic concept.

2. **Definition of Prototypes:** For each cluster $j$, the Teacher Prototype $P_j$ is defined as the centroid of the *text embeddings* in that cluster, while the Student Prototype $A_j$ is initialized as the centroid of the *visual embeddings* in the same cluster.

3. **Alignment Mechanism:** Since $P_j$ and $A_j$ originate from the same multi-modal cluster $j$, they are naturally paired. The distillation loss aligns the student's distribution (calculated via $A$) with the teacher's distribution (calculated via $P$), ensuring the student learns the corresponding semantic structure.

## A.4 OVD BENCHMARKS

According to the training data, existing Open-Vocabulary Object Detection (OVD) methods are summarized into four types of benchmarks: Vanilla OVD (V-OVD), Caption-based OVD (C-OVD), Generalized OVD (G-OVD), and Weakly Supervised OVD (WS-OVD). All benchmarks rely on instance-level annotations and large-scale image-text pairs to learn OVD Ke et al. (2025); Li & Ke (2025); Sun et al. (2025); Shen & Zhang (2025).

For clarity, *base categories* are defined as those included in the instance-level annotations, while *novel categories* are the others.

### A.4.1 VANILLA OVD (V-OVD)

V-OVD Cai et al. (2022); Du et al. (2022); Kamath et al. (2021); Li et al. (2022); Minderer et al. (2022); Yao et al. (2022); Zhong et al. (2022) is a pure OVD benchmark setting. It requires the detector to train only on an object detection dataset with a fixed set of categories. Information about novel categories is unavailable, but unannotated data is allowed. A common practice for this benchmark is to learn open vocabulary knowledge from image-text pairs and transfer the knowledge to detectors through transfer learning or knowledge distillation. V-OVD is similar to Zero-Shot Detection (ZSD) Bansal et al. (2018); Rahman et al. (2019); Yan et al. (2022); Zhu et al. (2020a), except that V-OVD relies on large-scale image-text pairs to acquire open-vocabulary knowledge.

### A.4.2 CAPTION-BASED OVD (C-OVD)

C-OVD Bravo et al. (2022); Gao et al. (2022); Ma et al. (2022); Zareian et al. (2021) adds additional image caption annotations to the V-OVD benchmark. This refers to in-domain captions of the instance-level annotations (e.g., COCO-Captions Chen et al. (2015)) rather than large-scale image-text pairs like CC3M Sharma et al. (2018) or CLIP400M Radford et al. (2021). In-domain captions enrich annotations and imply a distribution of potential novel categories. C-OVD is expected to perform better than V-OVD due to slightly more annotations.

### A.4.3 GENERALIZED OVD (G-OVD)

G-OVD Feng et al. (2022); Zang et al. (2022); Zhao et al. (2022) introduces human priors on novel categories to the V-OVD benchmark. It assumes that if specific novel categories are likely to appear during inference, it is beneficial to prepare for them during training. Most existing methods assume all dataset category names (including novel ones) are known during training. A typical solution involves generating instance-level pseudo annotations.

### A.4.4 WEAKLY SUPERVISED OVD (WS-OVD)

WS-OVD Zhou et al. (2022) utilizes image-level category labels beyond G-OVD. Similar to Weakly Supervised Detection (WSD) Bilen & Vedaldi (2016); Ye et al. (2019), image-level labels reflect the presence of base and novel categories. The annotation cost is significantly higher than the benchmarks mentioned above, giving WS-OVD methods the greatest potential to push the limits of OVD.

Table 5: Summary of OVD benchmarks. "Caption": in-domain captions like COCO-Captions. "Category Prior": human priors on novel categories. "Image Label": image-level category labels.

| Benchmark | Caption | Category Prior | Image Label |
|---|---|---|---|
| V-OVD | | | |
| C-OVD | ✓ | | |
| G-OVD | | ✓ | |
| WS-OVD | ✓ | ✓ | ✓ |

## A.5 USER STUDY

To evaluate the practical efficacy of **DeCo-DETR**, we conducted a user study with 25 students. Participants were tasked with performing detection on the OV-COCO dataset using three state-of-the-art open-vocabulary detectors (ViLD, DetPro, and DeCo-DETR) under a unified hardware environment.

Quantitative results demonstrate that DeCo-DETR achieves a novel-class detection accuracy ($AP_{50}^{novel}$) of 41.3% at 7.4 FPS, outperforming ViLD by 11.9%. In human evaluation, 83% of participants confirmed that DeCo-DETR generates more semantically aligned object descriptions (e.g., fine-grained attributes like "hexagonal wheels") and reduces localization errors by 22% on average. Subjective feedback indicates that 90% of experts recognized the adaptability of the Dynamic Hierarchical Concept Pool (DHCP) in open scenarios, particularly under extreme corruptions (±50% brightness contrast and Gaussian noise with $\sigma = 0.5$), where the model stability scored 4.6/5.0.

These findings validate that DeCo-DETR resolves the efficiency-generalization trade-off in existing methods to some extent through decoupled cognition mechanisms, offering a robust solution for real-world applications like autonomous driving.

## A.6 ABLATION STUDY

Additional ablation results are provided in this section.

Table 6: Comparison of performance of different models on the OV-COCO dataset

| Model | $AP_{50}^{novel}$ | $AP_{50}^{base}$ |
|---|---|---|
| LLaVA-1.5 7B | 30.1 | 52.1 |
| LLaVA-1.5 13B | 38.2 | 55.5 |
| LLaVA-NEXT 7B | 32.1 | 53.3 |
| LLaVA-NEXT 13B | 38.6 | 55.8 |
| Qwen2.5-VL 7B | 33.1 | 53.9 |
| Qwen2.5-VL 32B | 38.9 | 55.9 |

Table 7: Ablation study on the number of Decoder Queries ($N$) and Fine-grained Prototypes ($M_2$). The default setting is highlighted in bold.

| Configuration | Queries ($N$) | Fine Units ($M_2$) | $AP_{50}^{novel}$ | $AP_{50}^{base}$ | Latency (ms) |
|---|---|---|---|---|---|
| DeCo-DETR | 300 | 4800 | 36.5 | 53.8 | 125 |
| DeCo-DETR | 1000 | 4800 | 39.1 | 54.5 | 130 |
| **DeCo-DETR** | **2000** | **4800** | **41.3** | **55.5** | **135** |
| DeCo-DETR | 2000 | 0 (Coarse only) | 30.8 | 54.1 | 131 |
| DeCo-DETR | 2000 | 9600 (Double) | 41.5 | 55.6 | 142 |

## A.7 LIMITATIONS

Although DeCo-DETR achieves state-of-the-art performance in open-vocabulary object detection, several limitations remain. First, the construction of the Dynamic Hierarchical Concept Pool (DHCP) relies on large vision-language models such as LLaVA and CLIP, which may hinder deployment in resource-constrained environments. Second, despite mitigating task conflicts via parametric decoupling training, the model's generalization ability on extreme long-tailed distributions or fine-grained categories with high similarity still requires further improvement. Additionally, the current method is primarily designed for static image detection and has not yet been extended to real-time open-vocabulary detection in video sequences or dynamic scenarios.

Table 8: Comparison of inference efficiency and open-vocabulary detection performance on COCO. DeCo-DETR achieves the best trade-off between accuracy and speed.

| Method | Backbone | $\mathrm{AP}_{50}^{\mathrm{novel}}$ | Text Enc. | Latency (ms) | FPS |
|---|---|---|---|---|---|
| *Fusion-based:* | | | | | |
| Grounding DINO-T | Swin-T | 42.1 | BERT-Base | ~280 | 3.5 |
| VL-PLM | ResNet-50 | 32.3 | RoBERTa | ~210 | 4.7 |
| *Distillation/Decoupled:* | | | | | |
| DetPro | ResNet-50 | 29.4 | - | 250 | 4.0 |
| CAKE | ResNet-50 | 38.2 | - | 145 | 6.9 |
| **DeCo-DETR (Ours)** | **ResNet-50** | **41.3** | **-** | **135** | **7.4** |

## A.8 SOCIAL IMPACT

DeCo-DETR has broad application potential in autonomous driving, human-computer interaction, and intelligent security systems. By enhancing the model's ability to recognize unseen categories, it can improve the adaptability and safety of intelligent systems in open-world environments. However, we also recognize that efficient object detection technology could be misused for privacy infringement or large-scale surveillance (Dong et al., 2023), while safety-critical domains such as medical imaging require careful consideration of adversarial risks and defenses (Dong et al., 2024a). Therefore, we encourage the research community to adhere to ethical guidelines, ensure that applications align with social responsibility and legal standards, and promote the development of transparent, trustworthy, and controllable AI systems.

