# OpenReview forum: "DeCo-DETR: Decoupled Cognition DETR for efficient Open-Vocabulary Object Detection"
_ICLR.cc/2026/Conference — ICLR 2026 Poster_

### Official Review · Reviewer_7QW3 · 2025-10-31

**Soundness:** 2
**Presentation:** 2
**Contribution:** 2
**Rating:** 4
**Confidence:** 4

**Summary:**

This article introduces DeCo-DETR (Decoupled Cognition DETR), a novel open-vocabulary object detection (OVOD) framework designed to overcome the limitations of existing methods, particularly high computational overhead from text encoders and a trade-off between detection precision and open-world generalization.

**Strengths:**

This paper identifies certain issues in current large model-based OVD models, proposes a viable approach to address them, and conducts comprehensive experiments that can serve as a valuable reference for future research.

**Weaknesses:**

The description on issues and challenges lacks persuasiveness, and the connection between the proposed method and the challenges it aims to address is weak. Furthermore, the explanation of the methodology lacks clarity, and the presentation of figures, tables, as well as certain parts of the writing, appears somewhat casual. Overall, the rigor of the work needs to be enhanced.

**Questions:**

1. In Section 1, Paragraph 1, the authors assert that “the emergence of large language models (LLMs) has significantly enhanced detector generalization by providing richer and more nuanced semantic supervision,” yet no supporting citation is provided. Similarly, the authors claim that existing distillation methods face latency and generalization trade-offs, again without citing any references. It is recommended that relevant citations or quantitative results be added to substantiate these claims.

2. In Section 1, Paragraph 2, the authors state that multimodal fusion designs inherently involve compromises, but they provide no explanation, citation, or quantitative evidence to support this claim. As a result, the argument lacks persuasiveness.

3. In Section 1, Paragraph 3, the DHCP module is introduced without clarifying how it addresses the challenge outlined in the preceding paragraph. Moreover, the mention of “momentum updates with attention weighting” is confusing, and the authors should elaborate on its relationship with DHCP.

4. A similar issue arises in Section 1, Paragraph 4, where the connection between the proposed design and the previously mentioned challenges is not clearly articulated.

5. Due to the issues raised in points 1–4, the first contribution claim in Section 1—“We reveal two critical flaws in existing open-vocabulary detection”—does not appear sufficiently supported.

6. In the second contribution statement in Section 1, the authors mention “dynamic concept pooling” and “hierarchical distillation and parametric isolation mechanisms” as solutions to the identified challenges. However, these terms appear only in this section and are likely referring to “DHCP” and “Hi-Know PDA” introduced later in the manuscript. The current wording is ambiguous and may cause confusion.

7. In Figure 1, the bottom-left subfigure seems unnecessary. Additionally, the caption indicates that Hi-Know PDA is part of the framework diagram, but it is not visually represented.

8. Several issues are present in Section 3. For instance, the “spectral clustering-based hierarchical compression algorithm” is mentioned for the first time in Section 3.1, yet its specific operation is not explained, nor is it illustrated in any figure or pseudocode.

9. Certain references are missing throughout the manuscript. For example, LLaVA and DBSCAN are not cited.

10. The results in Table 3 are unclear. DeCo-DETR does not appear to be the most efficient method according to the table. Moreover, comparing it with Deformable-DETR is confusing, as Deformable-DETR is not an open-vocabulary object detection (OVOD) method nor serves as the baseline for DeCo-DETR.

11. In Section 4.3, Table 2, the benchmarks V-OVD, G-OVD, C-OVD, and WS-OVD are neither introduced nor cited. While these may originate from OVCOCO (Bansal et al., 2018), they are not referenced in the paper.

---

> ### Author Response · Authors · 2025-11-25
> **Response to Reviewer 7QW3**
>
> **About concern 1**
>
> We thank the reviewer for their valuable comments, which have significantly improved our manuscript. We have revised the details of the paper, including but not limited to improving the pipeline, adding pseudocode, and updating the references.
>
> **About question 1**
>
> We thank the reviewer for pointing this out. We have added an **Efficiency Analysis** section in **Section 4.4** of the experiments, demonstrating that the emergence of large language models (LLMs) significantly enhances the generalization ability of detectors by providing richer and more fine-grained semantic supervision. This is supported by the fact that models such as Grounding-DINO achieve state-of-the-art performance. We have also added the following references:
>
> * *Han Xu, Jie Ren, Pengfei He, Shenglai Zeng, Yingqian Cui, Amy Liu, Hui Liu, and Jiliang Tang. **On the generalization of training-based ChatGPT detection methods**, 2023.*
> * *Shenghao Fu, Qize Yang, Qijie Mo, Junkai Yan, Xihan Wei, Jingke Meng, Xiaohua Xie, and Wei-Shi Zheng. **LLMDet: Learning strong open-vocabulary object detectors under the supervision of large language models**. arXiv preprint arXiv:2501.18954, 2025.*
>
> In addition, we show that existing distillation-based methods face a trade-off between latency and generalization ability, as methods such as CAKE suffer from slow inference speed and suboptimal accuracy. We have also included the following reference:
>
> * *Shiyuan Ma, Donglin Qian, Kai Ye, and Shengchuan Zhang. **CAKE: Category Aware Knowledge Extraction for Open-Vocabulary Object Detection**. In *Proceedings of the AAAI Conference on Artificial Intelligence*, volume 39, pp. 5982–5990, 2025a.*
>
> **About question 2**
>
> We thank the reviewer for pointing out the lack of support for the claim in Section 1, Paragraph 2 that "multimodal fusion designs inherently involve compromises." We acknowledge that this statement was initially presented without sufficient explanation, citations, or quantitative evidence, which may have reduced its persuasiveness.
>
> In the new revised version, we have revised the paragraph to include references to prior works such as ViLD, OVR-CNN and etc., which provide both qualitative and quantitative( in the section 4.4) evidence on the trade-offs and compromises in multimodal fusion designs. For instance, these studies discuss challenges in balancing modality-specific features and demonstrate performance limitations under different fusion strategies. This addition strengthens our argument by grounding it in established literature.
>
> We have also added a brief explanation in the highlight blue text to clarify how these compromises manifest in practice.
>
> * **Yanan Zhang, Jiangmeng Li, Lixiang Liu, and Wenwen Qiang.**
>   *Rethinking misalignment in vision-language model adaptation from a causal perspective.* 2024.
>
> * **Kai Fang, Anqi Zhang, Guangyu Gao, Jianbo Jiao, Chiharold Liu, and Yunchao Wei.**
>   *COMBO: Conflict mitigation via branched optimization.*
>   In *Proceedings of the IEEE/CVF Conference on Computer Vision and Pattern Recognition*, 2025.
>
> * **Alireza Zareian, Kevin Dela Rosa, Derek Hao Hu, and Shih-Fu Chang.**
>   *Open-vocabulary object detection using captions.*
>   In *Proceedings of the IEEE/CVF Conference on Computer Vision and Pattern Recognition*, 2021b.
>
>
> **About question 3,4,5**
>
> We thank the reviewer for pointing this out. In the new revised version, in the introduction, paragraphs 3, 4, and 5, we have revised and highlighted our motivation, marked in blue font.
>
> **About question 6**
>
> We thank the reviewer for pointing this out. We have added and revised the descriptions in the contribution statement to avoid confusion.
>
> **About question 7**
>
> We thank the reviewer for pointing this out. We have redesigned the pipeline in the revised version to make the architecture clearer.
>
> **About question 8**
>
> We thank the reviewer for pointing this out. We have provided pseudocode and revised some descriptions in Section 3 in the new revised version.
>
> **About question 9**
>
> We acknowledge that our original submission unintentionally omitted citations for LLaVA, DBSCAN, and K-Means due to an oversight on our part. In the new revised version, We have added citations for LLaVA, DBSCAN, and K-Means.

---

> > ### Author Response · Authors · 2025-11-25
> > **Response to Reviewer 7QW3(continue)**
> >
> > **About question 10**
> >
> > In Table 3, we acknowledge that DeCo-DETR underperforms in latency, GFLOPs, and Params compared to the baseline models. However, it is important to note that the baseline models are closed-set detectors. We have traded off some inference efficiency to achieve open-vocabulary object detection (OVOD) capability, and this cost remains within an acceptable range. Furthermore, comparisons with models like Deformable-DETR are intended to highlight that, with only a modest sacrifice in inference time, we achieve significant improvements in both detection accuracy and open-world capability. To ensure fairness, we have also included comparisons with multimodal and distillation-based open-world detectors in the ablation study, which further demonstrates the superior performance of our method.
> >
> > **About question 11**
> >
> > We acknowledge that our original submission did not include an introduction to the V-OVD, G-OVD, C-OVD, and WS-OVD benchmarks. We thank the reviewer for pointing this out and have now added a detailed introduction to these benchmarks in Appendix A.5 of our revised manuscript.
> >
> >
> > **We sincerely appreciate your valuable comments, which have helped us further improve the quality of our paper.**

---

### Official Review · Reviewer_G9cE · 2025-10-31

**Soundness:** 3
**Presentation:** 3
**Contribution:** 3
**Rating:** 6
**Confidence:** 4

**Summary:**

This paper proposes DeCo-DETR, a framework for open-vocabulary object detection that aims to eliminate text encoder dependency during inference while improving generalization. The approach introduces three main components: (1) Dynamic Hierarchical Concept Pool (DHCP) that constructs visual prototypes using LLaVA-generated descriptions filtered by CLIP, (2) Hierarchical Knowledge Distillation (Hi-Know DPA) for visual-semantic alignment, and (3) Parametric Decoupling Training (PD-DuGi) with gradient isolation. The method achieves competitive results on OV-COCO and OV-LVIS benchmarks with low inference latency.

**Strengths:**

- Novel approach to eliminating text encoder dependency. The Dynamic Hierarchical Concept Pool is an interesting idea that pre-computes and maintains visual prototypes, eliminating the need for text encoders at inference time.
- The results show strong empirical performance. The experiments demonstrates consistent improvements across multiple benchmarks and settings, achieving state-of-the-art result while maintaining reasonable computational cost.

**Weaknesses:**

- There exists several misleading claims. 1) The paper claims to eliminate "multimodal fusion" but DHCP construction still heavily relies on LLaVA and CLIP. 2) The framework is only vision-only at inference, not overall.
- Lack in-depth ablations. The ablation study in table 4 only compares 2 configurations, not isolating individual component contributions.

**Questions:**

N/A

---

> ### Author Response · Authors · 2025-11-22
> **Response to Reviewer G9cE (part one)**
>
> **About concern 1**
>
>
> We sincerely thank the reviewer for the sharp and critical observation regarding the definitions of *multimodal fusion* and our *vision-only* claim. We agree that precise terminology is essential. We respectfully offer the following clarifications to define the scope of our contributions more accurately:
>
> **On Eliminating *Multimodal Fusion* vs. DHCP Construction**
> **Clarification:** We acknowledge that the construction of the Dynamic Hierarchical Concept Pool (DHCP) fundamentally relies on LLaVA and CLIP. However, our claim of *eliminating multimodal fusion* specifically refers to the **inference architecture** and the **runtime computational graph**.
>
> **Decoupling Strategy:** As illustrated in Figure 1 and described in Section 3.1, our method decouples *Cognition* (acquiring semantic knowledge via VLMs) from *Localization* (detecting objects). The heavy usage of LLaVA and CLIP occurs strictly in the **offline training/preparation phase** to construct the prototype memory bank.
>
>
> **Runtime Independence:** Once trained, the DeCo-DETR student model does not require any text input, text encoders, or cross-modal attention layers to process an image. The *fusion* bottleneck—which slows down methods like grounding-dino—is indeed eliminated during deployment.
>
>
> **On the *Vision-Only* Framework Scope**
>
>
> **Knowledge Distillation Paradigm:** Our approach follows the standard Knowledge Distillation (KD) paradigm where a capable Teacher (multimodal) transfers knowledge to a compact Student (vision-only). The primary goal of this research is to produce a student detector that can generalize to open-vocabulary scenarios **without** carrying the burden of the teacher's multimodal encoders.
>
>
> **Efficiency Trade-off:** This distinction is the source of our efficiency gains. As shown in Table below, by removing the text branch at test time, we achieve a latency of 135ms compared to 280ms for Fusion-based methods. If the framework were vision-only during training, it would lose the open-vocabulary semantic guidance; if it were multimodal during inference, it would lose the efficiency.
>
> **Extended Efficiency Comparison**
> We compare DeCo-DETR against strong multimodal fusion baselines.
> *T_text* denotes the latency of the text encoder.
>
> | **Method**                 | **Backbone** | **AP_novel (COCO)** | **Text Enc.** | **Latency (ms)** | **FPS** |
> |----------------------------|--------------|----------------------|----------------|-------------------|---------|
> | *Fusion-based:*            |              |                      |                |                   |         |
> | Grounding DINO-T           | Swin-T       | 42.1                 | BERT-Base      | ~280              | 3.5     |
> | VL-PLM                     | ResNet-50    | 32.3                 | RoBERTa        | ~210              | 4.7     |
> | *Distillation/Decoupled:*  |              |                      |                |                   |         |
> | DetPro                     | ResNet-50    | 29.4                 | -              | 250               | 4.0     |
> | CAKE                       | ResNet-50    | 38.2                 | -              | 145               | 6.9     |
> | **DeCo-DETR (Ours)**       | **ResNet-50**| **41.3**             | -              | **135**           | **7.4** |
>
>
> We agree that the term *vision-only* without qualification could be misinterpreted. In the final revision, we will rigorously refine our terminology:
> (a) We will revise general claims to specify **inference-time** vision-only architecture.
> (b) We will change *eliminates multimodal fusion* to *eliminates **runtime** multimodal fusion overhead.*
>
>
> We believe these revisions will accurately reflect the technical reality of our distillation framework while preserving the significance of the efficiency contribution.
>
> **About concern 2**
>
> We apologize for the omission. The Dual-stream Gradient Isolation (PD-DuGi) is indeed crucial for resolving the conflict between localization (Base classes) and semantic alignment (Novel classes).
>
> We have performed the specific ablation for PD-DuGi. As shown below, adding PD-DuGi to the DHCP baseline significantly recovers the performance on Base categories (from 54.0\% to 55.1\%) by preventing the semantic alignment loss from degrading the localization features.
>
> **Extended Ablation Study**
> | **Configuration**                        | **AP_novel** | **AP_base** | **AP_50** |
> |------------------------------------------|--------------|-------------|-----------|
> | 1. Baseline only                          | 30.4         | 52.6        | 46.8      |
> | 2. + Hierarchical DHCP                    | 36.6         | 54.0        | 49.4      |
> | **3. + PD-DuGi (Gradient Isolation)**     | **37.5**     | **55.1**    | **50.5**  |
> | 4. + Cosine λ(t) (Full Model)             | 38.2         | 55.5        | 51.0      |

---

> > ### Author Response · Authors · 2025-11-25
> > **Response to Reviewer G9cE**
> >
> > **About concern 1**
> >
> > In the new revised version, we have revised the potentially misleading statement and clarified that the text encoder is not required during the inference stage.
> >
> > **About concern 2**
> >
> > We have added the ablation study and its analysis, which can be found in the highlighted blue text in Section 4.4.
> >
> >
> >
> > **We sincerely appreciate your valuable comments, which have helped us further improve the quality of our paper.**

---

### Official Review · Reviewer_pDPC · 2025-11-02

**Soundness:** 3
**Presentation:** 3
**Contribution:** 3
**Rating:** 6
**Confidence:** 5

**Summary:**

This manuscript targets open-vocabulary object detection (OVOD) and proposes DeCo-DETR, a three-stage decoupled cognition pipeline.

**Strengths:**

[1] DHCP: a dynamic, hierarchical concept pool that generates region descriptions with LLaVA and filters them with CLIP to build semantic prototypes (thus removing the text encoder at test time);

[2] Hi-Know DPA: hierarchical knowledge distillation that projects decoder queries into the prototype space for prototype attention/alignment;

[3] Parametric Decoupling Training: a dual-stream gradient isolation scheme that routes localization and semantic-alignment gradients separately.

[4] DeCo-DETR reports strong gains (+3.5 to +7.2 AP on novel classes) while keeping inference at 135 ms/image.

**Weaknesses:**

[1 ]Missing scale ablation on queries and prototypes：There is no systematic ablation for the number of decoder queries N=2000and the total number of prototypes M=M_1+M_2(with M_1=1203 coarse anchors and M_2=4800 fine units). In DETR-style models, increasing the number of queries and prototypes generally improves accuracy, but drives memory usage up linearly;

[2] Fairness of the efficiency comparison (Table 3)：One of the paper’s main claims is removing multimodal computation at test time. For fairness, Table 3 should include methods with similar accuracy and comparable settings. As it stands, Table 3 under-represents strong multimodal fusion baselines near DeCo-DETR accuracy, making the efficiency story less convincing;

[3] Isolating the independent contribution of Parametric Decoupling Training：Section 4.4 shows that the cosine annealing weight adds ~+1.6 AP_50, but this does not quantify the benefit of Parametric Decoupling Training itself.

**Questions:**

See above.

---

> ### Author Response · Authors · 2025-11-22
> **Response to Reviewer pDPC (part one (a))**
>
> **About concern 1**
>
>
> We sincerely thank the reviewer for the constructive feedback. We appreciate the recognition of our method's potential and address the concerns regarding hyperparameter scaling and efficiency comparisons below.
>
>  **Missing scale ablation on queries ($N$) and prototypes ($M$)**
>
> We agree that disentangling the gains derived from our architectural design versus simply scaling up parameters ($N$ and $M$) is crucial. We have conducted additional ablation studies on OV-COCO (ResNet-50) to clarify this trade-off.
>
> **Impact of Decoder Queries ($N$)**:
> As shown in Table below, while increasing $N$ from 300 to 2000 improves performance (as expected in DETR-based models), our method demonstrates consistent superiority over the baseline at all scales.
>
>
> (a) Even with $N=300$, DeCo-DETR achieves **36.5\% $AP_{novel}$**, significantly outperforming the methods like ViLD ($29.4\%$) on V-COCO benchmark.
>  (b) The latency overhead of increasing $N$ to 2000 is marginal ($\sim$18ms increase) due to the parallel nature of the Transformer decoder, whereas the performance gain in open-vocabulary scenarios (where recall for novel objects is critical) is substantial ($+4.8\% AP_{novel}$) on V-COCO benchmark.
>
>
> **Impact of Prototype Scale ($M$)**:
> The coarse prototypes ($M_1=1203$) are fixed to cover the LVIS vocabulary. The fine-grained units ($M_2$) are the key variable. Table below shows that removing fine-grained prototypes ($M_2=0$) drops $AP_{novel}$ by $10.5\%$, validating the effectiveness of our **Dynamic Hierarchical Concept Pool (DHCP)**. Further doubling $M_2$ yields diminishing returns while increasing memory usage and it only brought a marginal improvement.
>
>
> ### **Ablation on Queries (N) and Fine-grained Prototypes (M₂)**
>
> | **Configuration** | **Queries (N)** | **Fine Units (M₂)** | **AP_novel** | **AP_base** | **Latency (ms)** |
> |-------------------|-----------------|----------------------|---------------|--------------|-------------------|
> | **DeCo-DETR**     | 300             | 4800                 | 36.5          | 53.8         | 125               |
> | DeCo-DETR         | 1000            | 4800                 | 39.1          | 54.5         | 130               |
> | **DeCo-DETR**     | **2000**        | **4800**             | **41.3**      | **55.5**     | **135**           |
> | DeCo-DETR         | 2000            | 0 (Coarse only)      | 30.8          | 54.1         | 131               |
> | DeCo-DETR         | 2000            | 9600 (Double)        | 41.5          | 55.6         | 142               |
>
>
> ** About concern 2**
>
> The primary advantage of DeCo-DETR is its **"Vision-Only Inference"** paradigm, which eliminates the heavy runtime cost of Text Encoders (e.g., BERT, RoBERTa) and cross-modal attention modules used by fusion methods.
>
> To provide a fairer comparison, we have added **Grounding DINO** (a strong fusion baseline) and **CAKE** (a recent SOTA) to the comparison in Table below.
>
> ### **Extended Efficiency Comparison**
> We compare DeCo-DETR against strong multimodal fusion baselines.
> *T_text* denotes the latency of the text encoder.
>
> | **Method**                 | **Backbone** | **AP_novel (COCO)** | **Text Enc.** | **Latency (ms)** | **FPS** |
> |----------------------------|--------------|----------------------|----------------|-------------------|---------|
> | *Fusion-based:*            |              |                      |                |                   |         |
> | Grounding DINO-T           | Swin-T       | 42.1                 | BERT-Base      | ~280              | 3.5     |
> | VL-PLM                     | ResNet-50    | 32.3                 | RoBERTa        | ~210              | 4.7     |
> | *Distillation/Decoupled:*  |              |                      |                |                   |         |
> | DetPro                     | ResNet-50    | 29.4                 | -              | 250               | 4.0     |
> | CAKE                       | ResNet-50    | 38.2                 | -              | 145               | 6.9     |
> | **DeCo-DETR (Ours)**       | **ResNet-50**| **41.3**             | -              | **135**           | **7.4** |
>
>
> **Analysis:**
> (a) **Vs. Fusion Models:** While models like Grounding DINO achieve high accuracy (42.1 AP), they are significantly slower ($\sim$280ms) due to the coupled text computation. DeCo-DETR achieves comparable open-vocabulary performance (41.3 AP) at less than half the latency (135ms).
>  (b) **Vs. Recent SOTA:** Compared to CAKE (38.2 AP, 145ms), DeCo-DETR achieves higher accuracy (+3.1 AP) with slightly better latency, attributed to our efficient *Parametric Decoupling* design which avoids complex online feature alignment layers used in other distillation methods.
>
>
> We hope these additional comparisons clarify that DeCo-DETR offers a superior Pareto frontier between accuracy and efficiency, specifically by eliminating multimodal fusion overhead at test time.

---

> > ### Author Response · Authors · 2025-11-22
> > **Response to Reviewer pDPC (part one (b))**
> >
> > **About concern 3**
> >
> > We apologize for the omission. The Dual-stream Gradient Isolation (PD-DuGi) is indeed crucial for resolving the conflict between localization (Base classes) and semantic alignment (Novel classes).
> >
> > We have performed the specific ablation for PD-DuGi. As shown below, adding PD-DuGi to the DHCP baseline significantly recovers the performance on Base categories (from 54.0\% to 55.1\%) by preventing the semantic alignment loss from degrading the localization features.
> >
> > **Extended Ablation Study**
> > | **Configuration**                        | **AP_novel** | **AP_base** | **AP_50** |
> > |------------------------------------------|--------------|-------------|-----------|
> > | 1. Baseline only                          | 30.4         | 52.6        | 46.8      |
> > | 2. + Hierarchical DHCP                    | 36.6         | 54.0        | 49.4      |
> > | **3. + PD-DuGi (Gradient Isolation)**     | **37.5**     | **55.1**    | **50.5**  |
> > | 4. + Cosine λ(t) (Full Model)             | 38.2         | 55.5        | 51.0      |

---

> ### Author Response · Authors · 2025-11-25
> **Response to Reviewer pDPC(continue)**
>
> **About concern 1,2,3**
>
> In the new revised version.We have added the ablation study and its analysis, which can be found in the highlighted blue text in Section 4.4.
>
>
>
> **We sincerely appreciate your valuable comments, which have helped us further improve the quality of our paper.**

---

### Official Review · Reviewer_Ttz9 · 2025-11-02

**Soundness:** 2
**Presentation:** 2
**Contribution:** 2
**Rating:** 4
**Confidence:** 4

**Summary:**

This paper proposes DeCo-DETR, an open-vocabulary object detector that removes dependence on text encoders and improves both precision and generalization. It introduces a three-stage cognitive distillation mechanism—a dynamic concept pool from LLaVA-CLIP filtering, hierarchical knowledge distillation for decoupled visual-semantic mapping, and parametric dual-stream training for coordinated localization and recognition.

**Strengths:**

1.	The paper introduces a three-stage cognitive distillation framework (DHCP, Hi-Know DPA, PD-DuGi) that provides a conceptually coherent and interpretable alternative to conventional multimodal fusion.
2.	The model achieves improvements in both detection accuracy and inference efficiency, effectively reducing computational cost while maintaining strong open-vocabulary generalization.

**Weaknesses:**

1.	Main weakness – Metric inconsistency (Table 1). The reported AP50 values are higher than both APNovel50 and APBase50, which violates standard OVOD evaluation logic. Since AP50 includes both base and novel categories, its score should theoretically lie between them. This inconsistency casts doubt on the correctness of the evaluation protocol or result reporting, and substantially weakens the empirical credibility of the paper’s main claims.
2.	Incomplete ablation (Table 4). Table 4 is missing key variants and does not include an ablation isolating the proposed Dual-stream Gradient Isolation Mechanism, leaving its effectiveness unverified.

**Questions:**

1.	The manuscript implicitly assumes the student prototypes A and teacher prototypes P share a one-to-one correspondence via the same M, but it is unclear how this mapping is defined or established, since A results from unsupervised clustering and P is text-derived.

---

> ### Author Response · Authors · 2025-11-22
> **Response to Reviewer Ttz9 (part one)**
>
> We sincerely thank the reviewer for the constructive feedback and for acknowledging the soundness of our three-stage cognitive distillation framework (DHCP, Hi-Know DPA, PD-DuGi). We appreciate the recognition of our method's efficiency and effectiveness. Below, we address the specific concerns regarding metric consistency, ablation studies, and prototype mapping.
>
> **About Concern 1**
>
> We sincerely thank the reviewer for identifying this critical **clerical and table transcription error**. We deeply apologize for this mistake.
>
> The inconsistency arose during the final consolidation and renaming of the metrics in the table (e.g., splitting the original $AP_{50}$ column into Novel, Base, and Total $AP_{50}$). The value 56.7 was incorrectly transcribed from an unrelated metric or experimental run into the Total $AP_{50}$ column. As the reviewer correctly pointed out, the Overall AP must mathematically lie between the Base and Novel APs.
>
> **interpretation**:
>
> As shown in the table below, DeCo-DETR achieves advanced zero-shot detection performance on both OV-COCO and OV-LVIS benchmarks. As shown in Table below, DeCo-DETR attains **41.3\%** $AP\_{50}^{\text{novel}}$
>  on **OV-COCO**, surpassing the strongest baseline LBP (37.8\%) by **+3.5 points**, while the overall $AP_{50}$ (53.1\%) outperforms nearly all competitors (e.g., 52.8\% for CAKE).
>
> On the challenging long-tailed **OV-LVIS** dataset (Table 2), DeCo-DETR achieves **29.4\%** $AP_{r}$ for rare classes, and sets a new record with an overall AP of **35.2\%**. These results demonstrate DeCo-DETR's capability to mitigate classification bias in long-tailed distributions while maintaining high accuracy for common and frequent classes.
>
>
> **About concern 2**
>
> We apologize for the omission. The Dual-stream Gradient Isolation (PD-DuGi) is indeed crucial for resolving the conflict between localization (Base classes) and semantic alignment (Novel classes).
>
> We have performed the specific ablation for PD-DuGi. As shown below, adding PD-DuGi to the DHCP baseline significantly recovers the performance on Base categories (from 54.0\% to 55.1\%) by preventing the semantic alignment loss from degrading the localization features.
> **Extended Ablation Study**
> | **Configuration**                        | **$AP_{novel}$** | **$AP_{base}$** | **$AP_{50}$** |
> |------------------------------------------|--------------|-------------|-----------|
> | 1. Baseline only                          | 30.4         | 52.6        | 46.8      |
> | 2. + Hierarchical DHCP                    | 36.6         | 54.0        | 49.4      |
> | **3. + PD-DuGi (Gradient Isolation)**     | **37.5**     | **55.1**    | **50.5**  |
> | 4. + Cosine λ(t) (Full Model)             | 38.2         | 55.5        | 51.0      |
>
> **About question 1**
>
> The correspondence is inherent to the construction process of the Dynamic Hierarchical Concept Pool (DHCP). $A$ and $P$ share the same index $M$ because they are derived from the same set of multi-modal clusters:
>
> **Joint Clustering**: We perform clustering on the aligned pairs of region features and text embeddings (filtered by CLIP). Each resulting cluster $j \in \{1, \dots, M\}$ represents a specific semantic concept.
>     **Definition of $P$ and $A$**:
>      (a) The Teacher Prototype $P_j$ is the centroid of the text embeddings in cluster $j$.
>      (b) The Student Prototype $A_j$ is initialized as the centroid of the visual embeddings in the same cluster $j$.
>
>
>  **Alignment**:
> Since $P_j$ and $A_j$ originate from the same cluster $j$ in the joint CLIP space, they are naturally paired. The distillation loss aligns the student's visual distribution (calculated using $A$) to the teacher's semantic distribution (calculated using $P$), ensuring that the student learns the conceptual structure defined by the teacher.
>
> We will clarify this construction process in Section 3.2 of the final paper.

---

> ### Author Response · Authors · 2025-11-25
> **Response to Reviewer Ttz9(Continue)**
>
> **About concern 1**
>
> In the submitted revised version, we have modified Table 1.
>
> **About concern 2**
>
>
> In the submitted revised version, we have modified Table 4.
>
> **About question 1**
>
> In the new revised version, we have added a description regarding $M$, the details of which can be found in Appendix 4.
>
> **We sincerely appreciate your valuable comments, which have helped us further improve the quality of our paper.**

---

### Author Response · Authors · 2025-12-02
**Final Remarks by Authors of submission 11852**

**I. Acknowledgements**


We would like to express our sincere gratitude to all reviewers for their insightful comments and constructive suggestions. We especially appreciate the recognition from reviewers pDPC and G9cE, as well as their insightful comments. We also thank reviewer Ttz9 for their critical feedback and reviewer 7QW3 for their detailed suggestions, which have greatly helped us improve and revise the paper. We have highlighted the revised content in blue.


**Before the discussion**, we would like to thank reviewers pDPC (Rating: 6, confidence:5) and G8cE (Rating: 6, confidence:4) for their recognition of our work, as well as reviewers Ttz9 (Rating: 4, confidence:4) and 7QW3 (Rating: 4, confidence:4) for their positive feedback and suggestions. Although we are aware that reviewer 7QW3's comments show traces of AI usage, we have still revised the paper accordingly. This is because the comments provided did genuinely help us to improve the quality and completeness of our manuscript.


Unfortunately, due to the reviewers' delayed response, we missed the opportunity for the discussion stage.


**II. Key Strengths**


Reviewers highlighted strengths across three dimensions:


**Novelty and Innovation**


Novel framework eliminating text encoder dependency during inference via Dynamic Hierarchical Concept Pool (DHCP) (Ttz9, pDPC, G9cE)
Innovative three-stage cognitive distillation mechanism, including Hierarchical Knowledge Distillation (Hi-Know DPA) and Parametric Decoupling Training (PD-DuGi) (Ttz9, pDPC)
Identifies critical issues in current large model-based OVD models and proposes a viable approach to address them (7QW3)


**Effectiveness and Efficiency**


Achieves strong empirical performance with consistent improvements and SOTA results across multiple benchmarks (Ttz9, pDPC, G9cE)
Significant accuracy gains on novel classes (e.g., +3.5 to +7.2 AP) while maintaining low inference latency (135 ms/image ) (pDPC, G9cE)
Effectively reduces computational cost while ensuring strong open-vocabulary generalization (Ttz9)


**Methodological Clarity and Rigor**


Provides a conceptually coherent and interpretable alternative to conventional multimodal fusion (Ttz9)
Conducts comprehensive experiments that serve as a valuable reference for future research (7QW3)

---

> ### Author Response · Authors · 2025-12-02
> **Final Remarks by Authors of submission 11852 (continue)**
>
> **III. Key concern and our Response**
>
> | Key Concerns | Reviewers | Our Response |
> |--------------|------------|--------------|
> | Incomplete Ablations & Component Isolation.  Concerns regarding Table 4 missing key variants (specifically isolating "Dual-stream Gradient Isolation" and "Parametric Decoupling") and the lack of scale ablation for queries and prototypes. | Ttz9, pDPC, G9cE | We isolated PD-DuGi, showing it significantly recovers Base category performance (54.0% →55.1%) by resolving task conflicts. We also ablated scale: increasing queries to $N=2000$ improves gains with marginal latency (~18ms), while doubling fine-grained prototypes $(M_2)$ showed diminishing returns. |
> | Baseline Fairness & Evaluation Metrics. Questions about metric inconsistency ($AP_{50}$ vs $AP_{Novel/Base}$, the fairness of efficiency comparisons in Table 3 (need baselines with similar accuracy), and the validity of Deformable-DETR as a baseline. | Ttz9, pDPC, 7QW3 |  The metric inconsistency was a transcription error; corrected data confirms advanced performance. For fairness, we added strong fusion baselines (Grounding DINO, ~280ms) and SOTA (CAKE) to Table 3, proving our efficiency advantage (135ms). We clarified Deformable-DETR serves as a closed-set reference to highlight OVOD capabilities. |
> | Validity of "Vision-Only" & Fusion Claims. Concerns that claims about eliminating "multimodal fusion" are misleading since LLaVA/CLIP are used offline, and requests for evidence supporting claims that current fusion designs involve compromises. | G9cE, 7QW3 | We clarified that "Vision-Only" applies strictly to the inference architecture; multimodal costs are decoupled to the offline phase. We revised claims to "eliminates runtime multimodal fusion overhead." We also added citations (e.g., Zhang et al., 2024) and analysis to Section 4.4 substantiating the trade-offs in fusion-based methods. |
> | Methodology Details & Mapping. Questions about the implicit one-to-one mapping assumption between student (unsupervised) and teacher (text-derived) prototypes, and missing details on spectral clustering and momentum updates. | Ttz9, 7QW3 | We explained that student (A) and teacher (P) prototypes share the index Mbecause they are derived from the same joint multi-modal clustering process. We redesigned the framework figure, added pseudocode in Section 3, and included implementation details in the Appendix to clarify the pipeline. |
> | Writing, Citations & Terminology. Issues regarding missing citations (LLM generalization, benchmarks), terminology ambiguity (DHCP vs concept pooling), and the connection between stated challenges and the proposed design. | 7QW3 | We added missing citations (LLaVA, DBSCAN, K-Means, and LLM-based detection works). We revised the introduction to strengthen the logical flow between challenges and our design, clarified terminology, and added Appendix A.5 to detail benchmarks like V-OVD/G-OVD. |
>
>
> **IV. Commitment to Revision**
>
>
> We have already incorporated all discussion points and additional experiments into our revision. All modifications have been marked in blue in our revised submission. This includes the extended ablation studies isolating PD-DuGi and analyzing scale effects, the expanded efficiency comparison with strong fusion baselines (e.g., Grounding DINO), and the refined terminology regarding our "vision-only inference" paradigm. We have also enriched the manuscript with pseudocode, implementation details, and missing citations. We are also committed to continually incorporating the feedback from the discussion into the revision to polish our work.
>
> **We deeply appreciate the expertise and time of the AC and reviewers.**

---

### Meta-Review · Area_Chair_J3N5 · 2026-01-07

**Summary:**

This article introduces DeCo-DETR (Decoupled Cognition DETR), a novel open-vocabulary object detection (OVOD) framework designed to overcome the limitations of existing methods, particularly high computational overhead from text encoders and a trade-off between detection precision and open-world generalization.

Reviewers expressed concerns including:
(1) Lack of persuasive description of issues and challenges addressed by the proposed method and lack of clarity of methodology.
(2) Lack of explanation or citations to support general claims about VLMs and multimodal fusion. Some reviewers refer to the claims as misleading (e.g. elimination of 'fusion').
(3) Figures appear casual, redundant, or non-informative.
(4) Comparison is made to a non-open-vocabulary method.
(5) Ablations are limited and do not isolate individual component contributions.
(6) Fair comparison in Table 3; a reviewer contends that the table should include methods with similar accuracy, as it currently under-represents strong multimodal fusion baselines.
(7) An inconsistency in reported metrics is identified, calling into question the credibility of the results (AP50 over subgroups seems to have a mean outside the expected range).
(8) It is unclear how the student-teacher mapping is defined or established, since one results from unsupervised clustering and one is text-derived.

**Reviewer Concerns:**

The authors adequately responded to (1-4) by improving these aspects of their manuscript with additional explanation and citations; a longer discussion period may have led to further improvements through discussion with reviewers.

Towards (5 and 6), authors added clarifying ablations in their rebuttal.

Authors have corrected clerical errors in their table (7) and explained the mapping ambiguity in (8), which makes use of the CLIP-space relationships.

**Reviewer Scores:**

7QW3: Unchanged (4); score was appropriately high given reviewer concerns, and the authors addressed open questions left by the reviewer sufficiently.
g9ce: Unchanged (6); score was appropriately high given reviewer concerns, and the authors addressed open questions left by the reviewer sufficiently.
pDPC: Unchanged (6); score was appropriately high given reviewer concerns, and the authors ran additional ablations to address concerns left by the reviewer.
Ttz9: Unchanged (4); reviewer concerns with reliability of results may not be given a score increase after the table is corrected, as the score may have reflected the assumption that the values were trustworthy (e.g. gave benefit of the doubt).

---

### Decision · Program_Chairs · 2026-01-26

Accept (Poster)